# Modeling of Yb:YAG Laser Beam Caustics and Thermal Phenomena in Laser–Arc Hybrid Welding Process with Phase Transformations in the Solid State

**DOI:** 10.3390/ma17102364

**Published:** 2024-05-15

**Authors:** Marcin Kubiak, Zbigniew Saternus, Tomasz Domański, Wiesława Piekarska

**Affiliations:** 1Faculty of Mechanical Engineering and Computer Science, Czestochowa University of Technology, Dabrowskiego 69, 42-201 Czestochowa, Poland; marcin.kubiak@pcz.pl (M.K.); tomasz.domanski@pcz.pl (T.D.); 2Faculty of Architecture, Civil Engineering and Applied Arts, Academy of Silesia, Rolna 43, 40-555 Katowice, Poland; wieslawa.piekarska@pcz.pl

**Keywords:** Yb:YAG laser hybrid welding, thermal phenomena, phase transformations

## Abstract

This paper focuses on the mathematical and numerical modeling of the electric arc + laser beam welding (HLAW) process using an innovative model of the Yb:YAG laser heat source. Laser energy distribution is measured experimentally using a UFF100 analyzer. The results of experimental research, including the beam profile and energetic characteristics of an electric arc, are used in the model. The laser beam description is based on geostatistical kriging interpolation, whereas the electric arc is modeled using Goldak’s distribution. Hybrid heat source models are used in numerical algorithms to analyze physical phenomena occurring in the laser–arc hybrid welding process. Thermal phenomena with fluid flow in the fusion zone (FZ) are described by continuum conservation equations. The kinetics of phase transformations in the solid state are determined using Johnson–Mehl–Avrami (JMA) and Koistinen–Marburger (KM) equations. A continuous cooling transformation (CCT) diagram is determined using interpolation functions and experimental research. An experimental dilatometric analysis for the chosen cooling rates is performed to define the start and final temperatures as well as the start and final times of phase transformations. Computer simulations of butt-welding of S355 steel are executed to describe temperature and melted material velocity profiles. The predicted FZ and heat-affected zone (HAZ) are compared to cross-sections of hybrid welded joints, performed using different laser beam focusing. The obtained results confirm the significant influence of the power distribution of the heat source and the laser beam focusing point on the temperature distribution and the characteristic zones of the joint.

## 1. Introduction

The most commonly used lasers in the industry are CO_2_ lasers and solid-state lasers. Solid-state disk-type YAG lasers with ytterbium-doped gain medium are currently the top advanced lasers, characterized by low optical quantum defects and high efficiency [1,2,3,4]. The Yb:YAG laser is used in the welding process as a single heat source as well as a hybrid heat source with an accompanying electric arc or a second laser beam (so-called dual beam processing) [5,6,7]. Both heat sources in the hybrid process usually cooperate in tandem with the leading laser beam or the leading electric arc.

HLAW is a popular welding technology due to its advantages in comparison to well-known electric arc welding or single-laser welding used in separate processes. Material deep penetration with a good gap filling is obtained when the laser beam and the electric arc cooperate in the process. In consequence, the improvement in weld quality is achieved by reducing the disadvantages of these two methods [1,8]. The HLAW technique allows for a better fit-up and reduces the distortions that appear when a material is heated to high temperatures. This is especially important in the welding process of closed profiles or long steel sheets [9,10,11,12,13].

Various process parameters determine the quality of joints welded classically by electric arc or laser beam, like the energy of the heat source and energy distribution in the material, welding speed, inclination angle of the torch, and many others [14,15,16,17,18]. Knowledge about thermal processes, including temperature fields, heating rates, and cooling rates, is crucial to predicting material behavior and allows the prediction of welding parameters as well as post-treatment processes in order to provide the joint’s desired shape and proper strength properties [11,12,16].

The amount and distribution of heat energy in the HLAW process determine the temperature field in the material. The coupled laser beam and electric arc define the heat transfer and motion of liquid steel in the FZ; as a result, the shape and size of the welded joint are determined [19,20,21,22]. An important issue in the HLAW model is the proper distribution of heat source power, reflecting the real process conditions as closely as possible. Mathematical models of heat flux distribution are constantly developed in the field of the theoretical study of laser-induced plasma formation [23,24], with some utilizing simplified models of Gaussian-like distributions [13,21,25,26,27,28]. Even then, some models describing the welding process can use simplified heat source distribution with mostly sophisticated models of plasma formation, droplet transfer, and free surface generation in the process [26,27,28].

Theoretical studies concerning the modeling of TEM_00_ laser beam heat sources usually take into account Gaussian models for lasing profiles [13,25]. However, prior works have shown the extensive difference between the real Yb:YAG laser profile and the theoretical Gaussian-like laser profile [18,29]. Previous works have also shown that the laser beam power distribution decreases with an increasing depth of material penetration as a function of the welding speed [21,23]. These facts contributed to the development of the HLAW heat source model in this study, based on kriging interpolation for the Yb:YAG laser beam and analytical models for the electric arc.

The knowledge about thermal phenomena and phase transformations in the solid state accompanying thermal processes is helpful in the estimation of many technological parameters that should be correctly set to ensure process stability and the best possible quality of steel products. The type of structure and the resulting properties of the heat-affected zone (HAZ) depend on the chemical composition of the steel and thermal cycle parameters such as heating rate, maximum heating temperature, residence time above the *A_c_*_3_ temperature, and cooling rate. Changes in the HAZ microstructure, including phase transformations and grain growth, are the cause of significant changes in the properties of this area compared to the base material. Different heating and cooling conditions during heat treatment contribute to the emergence of various structures in the HAZ, which leads to different mechanical properties [30,31,32,33,34,35].

This paper presents a 3D model of the thermal phenomena occurring during the HLAW process of butt joints made of S355 steel, with liquid material flow in the FZ taken into consideration. Numerical methods such as Chorin’s projection and the finite volume method (FVM) [36] are used to solve continuum mechanics equations for the determination of temperature distribution in the joint and fluid flow in the FZ. Numerical algorithms assume effective heat capacity, with the latent heat of fusion [13,18,20,25] and the latent heat of evaporation [24,25] taken into account. The major novelty in this work is the use of the geostatistical kriging interpolation method [18,37,38] to define the Yb:YAG laser beam heat source power distribution based on the real heat source profile measured experimentally. An electric arc source is modeled using double-ellipsoidal Goldak’s heat source power distribution [39,40]. Measurements of Yb:YAG laser intensity for different beam focusing conditions, as well as measurements of arc current and voltage, are taken into account in the heat source model. Computer simulations are partially verified by the results of real welding tests. The analysis of phase transformations in the solid state in the HAZ of the welded joint is carried out based on the supercooled austenite decay diagrams (welding CCT diagrams). The direct “in situ” method and the simulation dilatometric method are most often used to investigate the kinetics of phase changes under the conditions of welding thermal cycles. Elaborated theoretical models with comprehensive numerical solutions allowed the prediction of fusion zone geometry and heat-affected zone geometry in butt-welded plates made of S355 steel, as well as the structural composition of hybrid welded steel—the key quality parameters of the weld (the size, shape, and microstructure) for different process parameters.

## 2. Experimental Research

Our experimental research is performed at the Research Network ŁUKASIEWICZ—Welding Institute. Welding tests and measurements of the energetic properties of the laser beam and electric arc are carried out using a Trumpf D70 laser head equipped with the disk laser TruDisk 12002 (US. TRUMPF Inc., Farmington Industrial Park CT, 06032 Farmington, CN, USA), presented in Figure 1.

The laser’s maximum power is 12 kW. Fibers with LLK-D plugs are joined to the laser. The fibers have diameters of 200 μm, 300 μm, 400 μm, and 600 μm. A collimator lens (focal length of *f_c_* = 200 mm) and a focusing lens (focal length of *f* = 400 mm) are included in the head equipment. Changing pumping fibers adjusts the focusing diameter in this system. In our experimental research, a fiber with a diameter of *dLLK* = 400 μm is used. The diameter of the beam equals *d* = 0.8 mm, according to the relation *d* = *dLLK* × *f*/*f_c_*.

### 2.1. Laser Beam Profile

The Prometec UFF100 laser beam analysis system is used to determine the laser beam power density distribution and the caustics of the laser beam (Figure 2). The UFF100 is a diagnostic device for measuring and monitoring the laser beam of high-power lasers. It scans a laser beam through a measuring element—a hollow needle. On its surface, on the side of the incidence of the laser beam, a hole is drilled to allow the laser beam to enter the center of the needle. The rotating needle is introduced into the area of the laser beam. The laser beam enters the hollow needle and is transported to the detector. The rotation of the needle in the plane perpendicular to the axis of the laser beam enables it to be scanned along the trajectory of the needle opening. The precise needle-shifting system enables the beam to be shifted along its cross-section in several different cross-sectional planes.

The detector signals are converted into a digital signal and temporarily stored using a 16-bit processor integrated with the detector. The signals can be sent via a standard interface to a computer for processing and displaying the downloaded data. The appropriate program enables the three-dimensional imaging of the power density distribution and the two-dimensional profiling of the laser beam. In the measurement, a 30 µm pinhole diameter is used, and the resolution of the picture is the following: x-resolution = 81 and y-resolution = 41 pixels in a measurement window from −1 mm to 1 mm, respectively, in the x and y directions. The three-dimensional imaging of the percentage density distribution of the laser is presented in Figure 3.

Table 1 shows the measured distribution of the Yb:YAG laser beam (as percentages) for two different focusing conditions. Figure 4 illustrates laser beam caustics for radii *w_x_* and *w_y_* in the plane (*x*, *y*), according to norm PN-EN ISO 11146 [41].

### 2.2. Electric Arc Parameters

Electric arc current and voltage are constantly measured and presented on the machine monitor during the welding tests. Measurements are also written into a local database for further use. Figure 5 presents a picture made on the computer screen responsible for monitoring the current and voltage during the process. The energetic parameters of the arc are stored in the machine database and further transferred as digital data (as a time-parameter relationship) into the computer solver.

### 2.3. Welding Tests

The HLAW process is performed to compare the geometry of the FZ and the HAZ in welded joints with numerically simulated temperature distributions. Sheets made of S355 steel with dimensions of 250 × 50 × 5 mm are butt-welded using the Yb:YAG laser and GMAW. The energetic parameters used in the tests are shown in Table 2.

The two welding tests are performed for beam focusing at the top surface of the joint (z = 0) and for focusing with an offset of z = 5 mm from the top surface of the joint. Figure 6 presents the face of the weld and the cross-section of the joint welded for z = 0, whereas Figure 7 presents macroscopic pictures of the joint obtained for z = 5 mm.

## 3. Mathematical and Numerical Model

Heat transfer mostly depends on convection and conduction. In the HLAW process, two heat sources cooperate in a single welding pool [22,25]. Fluid flow in the fusion zone is generated by electromagnetic force, Marangoni shear stress, arc forces, recoil pressure, surface tension, buoyancy forces, etc. [26,27,28]. In this study, we consider the Marangoni effect with a surface tension diagram depending on temperature, in accordance with the research presented in [28,42,43], and buoyancy forces in the FZ. Material between the solidus and liquidus temperatures as a mixture of solid particles and liquid steel (the so-called mushy zone) is assumed to be a porous medium according to Darcy’s equations. Phase transformations occurring in the mushy zone and at high temperatures above the boiling point of steel are taken into account (solid–liquid and liquid–gas transformations). A three-dimensional model of the HLAW process is presented in Figure 8.

### 3.1. Governing Equations

Heat transfer and fluid flow are calculated by solving the equations of conservation of energy, momentum, and mass [26,34]:(1)$$\frac{\partial \rho }{\partial t}+\frac{\partial }{\partial x_{i}}\left(\rho v_{i}\right)=0$$
(2)$$\frac{\partial \left(\rho v_{i}\right)}{\partial t}+\frac{\partial }{\partial x_{j}}\left(\rho v_{i}v_{j}\right)=-\frac{\partial p}{\partial x_{i}}+\frac{\partial }{\partial x_{j}}\left(\mu \frac{\partial v_{i}}{\partial x_{j}}\right)+g\beta_{T}\left(T-T_{ref}\right)-\frac{\mu }{\rho K}v_{i}$$
(3)$$\frac{\partial }{\partial x_{i}}\left(\lambda \frac{\partial T}{\partial x_{i}}\right)=C_{ef}\left(\frac{\partial T}{\partial t}+v_{i}\frac{\partial T}{\partial x_{i}}\right)-\tilde{Q},$$
where ρ is the density, $\beta_{T}$ is the coefficient of thermal volume expansion, **g** is the acceleration of gravity, μ is the dynamic viscosity, $T_{ref}$ is the reference temperature, *T* = *T*(*x_i_*, *t*) is the temperature at point *x_i_*, *K* is the porous medium permeability, *v_i_* is the velocity vector, *C_ef_* = *C_ef_*(*T*) is the effective heat capacity, λ = λ(*T*) is the thermal conductivity, and $\tilde{Q}$ is the volumetric heat source for the HLAW process.

Equation (2) is completed by the initial condition t=0:v=0 and the boundary conditions defined at the FZ boundary:(4)$$\Gamma :\;v{|}_{\Gamma }=0,\;\;\tau_{s}\;=\mu \frac{\partial v}{\partial n}=\frac{\partial \gamma }{\partial T}\frac{\partial T}{\partial s},$$
where $\tau_{s}$ is the Marangoni stress tangent to the surface and γ is the coefficient of surface tension.

In this paper, the surface tension gradient is considered to vary with temperature according to [28,42,43].

The initial condition $t=0:T=T_{0}$ and the boundary conditions complete Equation (3), taking into account convection, radiation, and evaporation:(5)$$\Gamma :\;-\lambda \frac{\partial T}{\partial n}=\alpha (T{|}_{\Gamma }-T_{0})+\varepsilon \sigma (T^{4}{|}_{\Gamma }-T_{0}^{4})-q_{o}+q_{v},$$
where *α* is the coefficient of convection, *ε* is the coefficient of radiation, *σ* is the Stefan–Boltzmann constant, $q_{o}$ is the heat flux coming from the distribution of the hybrid heat source at the top surface of the joint (*z* = 0), $q_{v}$ is the heat flux due to material evaporation, and *Γ* is the boundary of the analyzed object [34].

The heat flux due to material evaporation depends on alloy constituents and can be defined as follows [35,44,45]:(6)$$q_{v}\;=q_{v}\left(x,y\right)=\sum_{i=1}^{n}J_{i}\left(T\right)H_{b}\left(i\right),$$
where *J_i_*(*T*) is the evaporation flux of the *i*-th alloy constituent [kg/m^2^s] and *H_b_*(*i*) is the latent heat of evaporation [J/kg]. 

Based on the data given in [35,44], a unified model of average evaporation flux is considered in the temperature range (*T_L_*; *T*_max_), where *T*_max_ is the maximum temperature of the thermal cycle and *T_L_* is the liquidus temperature:(7)$$\begin{array}{c} q_{v}\;=q_{v}\left(x,y\right)=J\left(T\right)H_{b},\\ J\left(T\right)=\left\{\begin{array}{cc} 8.7\times 10^{-5}T^{2}-0.32T+297 & for\;\;T^{*}\geq T\geq T_{L}\\ J\left(T^{*}\right)+J\left(T_{\max}\right)-J\left(T^{*}\right)\frac{T-T^{*}}{T_{\max}-T^{*}} & for\;\;T_{\max}\geq T\geq T^{*}, \end{array}\right. \end{array}$$
where the values of coefficients determining *J*(*T*) in the temperature range *T*^*^ ≥ *T* ≥ *T_L_* (*T*^*^ = 2800 K) are assumed for discrete values of *J* equal to 0, 50, and 80 kg/m^2^s at temperatures of 1800, 2600, and 2800 K. The value of *J*(*T*_max_) is 90 kg/m^2^s. The latent heat of evaporation (*H_b_*) is 76 × 10^5^ J/kg.

The Carman–Kozeny equation describes the permeability *K* of a porous medium in the mushy zone as follows:(8)$$K=K_{0}\frac{f_{l}^{3}}{{\left(1-f_{l}\right)}^{2}};\;\;\;\;\;K_{0}\;=\frac{d_{0}^{2}}{180}$$
where *d*_0_ is the average diameter of a solid particle and $f_{l}$ is the coefficient of porosity.

The capacity model [34] considers the latent heat of fusion and evaporation. A linear approximation of the solid fraction between the solidus and liquidus temperatures (Equation (9)) and the liquid fraction in the liquid–gas region (Equation (10)) are assumed.
(9)$$C_{ef}\left(T\right)=\rho_{SL}c_{SL}+\rho_{S}\frac{H_{L}}{T_{L}-T_{S}}\;\;\;\mathrm{for}\;\;\;T\in \left[T_{S};T_{L}\right]$$
(10)$$C_{ef}\left(T\right)=\rho_{L}c_{L}+\frac{\rho_{L}H_{b}}{T_{\max}-T_{b}}\;\;\;\mathrm{for}\;\;\;T\geq T_{b},$$
where *T_s_* is the solidus temperature, $c_{SL}\rho_{SL}=c_{S}\rho_{S}\left(1-f_{l}\right)+c_{L}\rho_{L}(f_{l})$ are the density and specific heat ratio, *H_L_* is the latent heat of fusion, and *T_b_* is the boiling temperature.

### 3.2. Hybrid Heat Source

The heat source power distribution is responsible for the temperature distribution in the material. Therefore, the description of the Yb:YAG laser beam caustics is performed using interpolation methods [18]. The kriging method at a point (x, y) consists of a linear combination of observations. The function of the weighted average provides the following estimate:(11)$$\tilde{f}\left(x,y\right)=\sum_{i=1}^{n}w_{i}f\left(x_{i},y_{i}\right),$$
where *w_i_* are weight coefficients assigned to observations, $f\left(x_{i},y_{i}\right)$ is the function’s real value at a point of measurement, and n is the count of sampling points within the circle of radius *r_k_*.

Coefficients *w_i_* are calculated in a system of equations:(12)$$\left[\begin{array}{cccccc} 0 & \gamma \left(h_{12}\right) & \gamma \left(h_{13}\right) & \ldots & \gamma \left(h_{1n}\right) & 1\\ \gamma \left(h_{21}\right) & 0 & \gamma \left(h_{23}\right) & \ldots & \gamma \left(h_{2n}\right) & 1\\ \gamma \left(h_{31}\right) & \gamma \left(h_{32}\right) & 0 & \ldots & \gamma \left(h_{3n}\right) & 1\\ \ldots & \ldots & \ldots & \ldots & \ldots & \ldots \\ \gamma \left(h_{n1}\right) & \gamma \left(h_{n2}\right) & \gamma \left(h_{n3}\right) & \ldots & 0 & 1\\ 1 & 1 & 1 & \ldots & 1 & 0 \end{array}\right]\;X\;\left[\begin{array}{c} w_{1}\\ w_{2}\\ w_{3}\\ \ldots \\ w_{n}\\ \lambda \end{array}\right]\;=\left[\begin{array}{c} \gamma \left(d_{1}\right)\\ \gamma \left(d_{2}\right)\\ \gamma \left(d_{3}\right)\\ \ldots \\ \gamma \left(d_{n}\right)\\ 1 \end{array}\right],$$
where λ is the Lagrange multiplier, $\gamma \left(h_{ij}\right)$ is the theoretical semivariogram at distance *h_ij_* between basic points, and $\gamma \left(d_{i}\right)$ is the theoretical semivariogram at distance *d_i_* between observations and basic points.

The theoretical semivariogram (functions $\gamma \left(h_{ij}\right)$ and $\gamma \left(d_{i}\right)$) is approximated by linear function $\gamma (h)=C_{0}+Sh$ tending to sill *S* as h -> ∞, in which *C*_0_ is the nugget effect (function of discontinuity). The solution to Equation (12) gives the heat source distribution in the interpolation mesh.

The penetration depth in the laser melting process of metals is about 10^−4^–10^−5^ cm. Absorbed laser light vaporizes the material and forms a “keyhole” that contains ionized vapor. The laser power is absorbed in the ionized vapor and transferred to the walls of the “keyhole”, forming the weld pool. The mechanism of material melting by a laser beam, considered in engineering practice as a simplified model without recoil pressure, is usually recognized as a volumetric heat source model [35,39,46,47]. In this work, the volumetric heat source power distribution is calculated assuming interpolated distribution in the radial direction, with the “keyhole” considered a truncated cone with a linear decrease in energy intensity with material penetration:(13)$$Q_{1}(x,y,z)=\eta Q_{S}\tilde{f}\left(x,y\right)\left(1-\frac{z}{s}\right),$$
where $Q_{S}=\alpha P/\pi \omega_{0}s$ is the laser power per unit area [W/m^2^], *P* is the pumped power [W], *ω*_0_ is the radius of the beam [m], *s* is the heat source penetration depth [m], *α* is the heat source coefficient (for the cone-like shape of the heat source volume, *α* = 3), and *η* is the efficiency (absorption coefficient).

Figure 9 shows a comparison between analytical Gaussian-like laser power distributions, described in detail in [35], interpolated by the kriging algorithm heat source power distribution [18] and the real distribution measured by UFF100. The interpolation of the heat source is made for a grid step of Δ*h* = 0.02 mm. The basis for interpolation and the analytical model are experimental data for beam focusing at z = 0. The comparison between the percentage power intensity distribution and the measured distribution of the Yb:YAG laser beam is illustrated in the central axes (x = 0 and y = 0). As presented in Figure 9, there is a visible difference in power distribution mapping between the analytical models and the results of the laser beam profiles obtained experimentally, whereas the interpolated model represents experimental data with appropriate accuracy, especially in the case of local extrema occurring in the heat source distribution.

Goldak’s heat source [39] is used to describe an electric arc heat source (Equation (14)). This ‘double-ellipsoidal’ model glues together two half-ellipses using a semi-axis (Figure 10).
(14)$$Q_{2}=\left\{\begin{array}{l} q_{1}(x,y,z)=\frac{6\sqrt{3}f_{1}Q_{A}}{abc_{1}\pi \sqrt{\pi }}\times \exp(-3\frac{x^{2}}{c_{1}^{2}})\times \exp(-3\frac{y^{2}}{a^{2}})\times \exp(-3\frac{z^{2}}{b^{2}})\;\;\;\;\mathrm{for}\;\;\;x<x_{o}\\ q_{2}(x,y,z)=\frac{6\sqrt{3}f_{2}Q_{A}}{abc_{2}\pi \sqrt{\pi }}\times \exp(-3\frac{x^{2}}{c_{2}^{2}})\times \exp(-3\frac{y^{2}}{a^{2}})\times \exp(-3\frac{z^{2}}{b^{2}})\;\;\;\mathrm{for}\;\;\;x\geq x_{o}, \end{array}\right.$$
where *f*_1_ and *f*_2_ (*f*_1_ + *f*_2_ = 2) are coefficients of heat source energy proportionality, and *a*, *b*, *c*_1_, *c*_2_ are axes of the front ellipsoid and the rear ellipsoid.

The resultant distribution of electric arc heat source energy is a product, described as *Q*_1_*(x,y,z)* = *q*_1_*(x,y,z)* + *q*_2_*(x,y,z)*, where $Q_{A}=\eta_{A}IU$ is the electric arc heat source power, *I* is the arc current, *U* is the arc voltage, and *η_A_* is the efficiency.

The laser–arc hybrid heat source distribution is considered using Goldak’s heat source and an interpolated laser beam heat source operating in tandem, in a distance *d*.

### 3.3. Numerical Solution

The differential equations of the phenomena are solved using Chorin’s projection and the finite volume method (FVM) [36]. To establish a numerical model of heat transfer in the welded workpiece and fluid flow in the welding pool, the following assumptions are made:○The flow is Newtonian, laminar, and incompressible;○The free surface is not considered;○The liquid material velocity field is mostly generated by the Marangoni effect and natural convection;○The interactions between the base material and the liquid material are neglected;○The evaporation is considered only in the energy equation;○Surface forces, like recoil pressure, are omitted in this study.

The authors developed a solver for the analysis of physical phenomena accompanying welding processes using the Object Pascal programming language. The solver is composed of two modules for estimating the velocity field in the melted zone, determined by the solidus temperature (*T_s_*) and the temperature field. A staggered grid is used in spatial discretization to ensure the stability of the algorithm in terms of the odd–even decoupling between velocity and pressure. The vectors of velocity are determined at the center of the faces of the control volume. Pressure, density, and temperature are calculated in the center of the control volume. Figure 11 presents the elementary control volume on a staggered grid.

Since the forward Euler integration scheme as well as central difference quotients are used in the calculations, the quality of the results depends on the conditions of stability. Therefore, the time-step constraints and stabilization methods for high Peclet numbers are taken into account in solution algorithms [36].

## 4. Phase Transformations in the Solid State

The presented model uses the intersection of the cooling curves with the interpolated welding CCT diagram to define the start and final times, as well as the start and final temperatures. The kinetics of phase transformations in the solid state are modeled using classical JMA and KM models.

### 4.1. Definition of Interpolated CCT Diagram

The start (*F_s_*, *P_s_*, *B_s_*, *M_s_*) and final temperatures (*F_f_*, *P_f_*, *B_f_*, *M_f_*) of each phase transformation and the final fractions of structural constituents during cooling are determined experimentally for S355 steel at chosen cooling rates, determined within the temperature range from 800 °C down to 500 °C. Based on the obtained results (Figure 12), interpolation functions are determined to define the CCT diagram (Figure 13). In the upper part of Figure 12 white dots corresponds to results of performed dilatometric analysis, whereas final fractions of phases, illustrated in the lower part of Figure 13 are determined based on microscopic analysis of microstructure composition of analyzed steel.

The interpolation functions used in our calculations are summarized in Table 3.

Volumetric fractions of diffusive phases (austenite, ferrite, pearlite, and bainite) are determined for heating (Equation (15)) and cooling (Equation (16)) processes using the JMA formula [32,35], as follows:(15)$${\tilde{\eta }}_{A}(T,t)=\eta_{b(.)}(1-\exp(-bt^{n})$$
(16)$$\begin{array}{l} \eta_{\left(\cdot \right)}\left(T,t\right)=\eta_{\left(\cdot \right)}^{\%}{\tilde{\eta }}_{A}\left(1-\exp\left(-b{\left(t(T)\right)}^{n}\right)\right),\;\\ \;\;\;\;{\underline{\eta }}_{A}-\sum_{k=1}^{4}\eta_{k}\geq 0,\;\;\;\;\;\sum_{k=1}^{5}\eta_{k}^{\%}=1, \end{array}$$
where *η_b_*_(.)_ is the sum of volumetric fractions of the base material structure (*η_b_*_(.)_ = 1); $\eta_{\left(\cdot \right)}^{\%}$ is the maximal phase fraction for the determined cooling rate, estimated based on the CCT diagram; ${\tilde{\eta }}_{A}$ is the austenite fraction formed due to heating; and $\eta_{k}$ is the phase fraction formed before the calculated phase transformation during cooling.

Coefficients *b* and *n* are determined by the starting (*η_s_* = 0.01) and final (*η_f_* = 0.99) conditions for the phase transformation as follows [35]:(17)$$\;b(T)=-\frac{\ln(\eta_{f})}{{\left(t_{s}\right)}^{n(T)}},\;\;\;\;n\left(T\right)=\frac{\ln\left(\ln\left(\eta_{f}\right)/\ln\left(\eta_{s}\right)\right)}{\ln\left(t_{s}/t_{f}\right)},$$
where *t* is the time, *t_s_ = t_s_*(*T_sA_*) and *t_f_ = t_f_*(*T_fA_*) are the phase transformation start and final times, and *T_sA_* and *T_fA_* are the start and final temperatures.

The volumetric fraction of martensite (*ηM*) during cooling is estimated using the KM equation [33,35]:(18)$$\eta_{M}\left(T\right)=\eta_{\left(\cdot \right)}^{\%}{\tilde{\eta }}_{A}\left(1-\exp\left(-k{\left(M_{s}-T\right)}^{m}\right)\right),\;\;\;T\in \left[M_{s,}M_{f}\right].$$

Coefficient *k* depends on the martensite phase start and final temperatures (*M_s_* and *M_f_*), which are also determined using an interpolated CCT diagram [35]:(19)$$k=-\frac{\ln(\eta_{S})}{M_{s}-M_{f}}=-\frac{\ln(0.01)}{M_{s}-M_{f}}.$$

### 4.2. Model of Isotropic Strain

Thermal and structural strain during the heating and cooling of welded steel is calculated as a solution to the increase in isotropic strain [31]:(20)$$d\varepsilon^{Tph}=\sum_{i=1}^{i=5}\left(\alpha_{i}\eta_{i}dT-\mathrm{sgn}\left(dT\right)\varepsilon_{i}^{ph}d\eta_{i}\right),$$
where $\alpha_{i}=\alpha_{i}\left(T\right)$ are the thermal expansion coefficients of austenite, bainite, ferrite, martensite, and pearlite; $\varepsilon_{i}^{Ph}=\varepsilon_{i}^{Ph}\left(T\right)$ is the isotropic structural strain resulting from the transformation of the base material into austenite during heating and each phase (ferrite, pearlite, bainite, or martensite) arising from austenite during cooling; $d\eta_{i}$ is the volumetric fractions of phases; and sgn(.) is a sign function.

## 5. Results and Discussion

Computer simulations were executed for flat sheets made of S355 steel with dimensions of 250 × 50 × 5 mm. A grid with a spatial step set to 0.02 mm was assumed (the same as the spatial step of the heat source interpolation grid). The thermo-physical properties used in the simulations are shown in Table 4.

### 5.1. Thermal Phenomena

Two simulations with different laser beam focusing conditions (z = 0, z = 5 mm) were performed. The technological parameters used in our experimental research are assumed in computer simulations of the hybrid welding process. The distributions of temperature presented in Figure 14 and Figure 15 show the differences in the geometry of the FZ (solid line) and the HAZ (dashed line) obtained in the simulations, whereas the velocity distributions show the differences in melted material flow in the fusion zone. There are also visible differences in the melted material’s velocity field. In the case of focusing at z = 5 mm, the maximal values of melted material velocity are about two times smaller than in the case of laser beam focusing at the top surface of the welded element.

The temperature distributions in the cross-section of the weld are presented in Figure 16 and Figure 17. Two sections are presented in these figures, with the FZ boundary marked by a solid line and the HAZ boundary marked by a dashed line. These two sections correspond to the deepest penetration made by the leading laser beam (x = 5.4 mm) and the widest FZ area generated by the following electric arc (x = 12 mm) in the tandem. The results of the temperature field for beam focusing at z = 5 mm show that only partial material melting is achieved (Figure 17).

The characteristic zones of hybrid welded joints (FZ and HAZ) in the cross-section were compared to macroscopic pictures of the weld in order to verify the correctness of the assumed mathematical models and numerical solutions. Figure 18 shows a macroscopic picture of the weld in the cross-section, with the imposed predicted boundary of the FZ (*T_S_* = 1750 K) marked as a solid line and the predicted boundary of the HAZ (*T_A_* = 1080 K) marked as a dashed line.

The algorithm estimating the geometry of the FZ and the HAZ assumes the verification of characteristic temperatures in the entire volume of the thermally influenced zone. In the case of the melted zone, the “widest” boundary isoline of the FZ is determined by the solidus temperature relative to the middle of the plane of heat source activity. In the case of the HAZ, it is an isoline of the austenitization temperature in the cooling process of the joint, oriented perpendicular to the direction of the weld line.

### 5.2. Phase Transformations in the Solid State

#### 5.2.1. Tuning of the Model

Calculations were conducted for a constant heating rate (*v_h_* = 100 K/s) and three different cooling rates (*v*_8/5_ = 2, 50, and 200 K/s). The same parameters were used in our experimental research, carried out on a thermal cycle simulator equipped with thermocouples and a dilatometer. The comparison between the calculations and the experiment allowed for the proper definition of thermal expansion coefficients as well as structural strains. Figure 19, Figure 20 and Figure 21 present the calculated kinetics of phase transformations in the solid state and isotropic strain for different cooling rates.

The right side of Figure 19, Figure 20 and Figure 21 shows the kinetics of phase transformations for heating and cooling. It is assumed, based on a microscopic analysis of welded steel, that the base material consists of 40% pearlite and 60% ferrite. During heating, up to austenitization temperatures, these fractions are transformed into austenite according to the kinetics described in Equation (15). During the cooling process, austenite is transformed into the final structure according to Equation (16) for diffusive transformations (ferrite, pearlite, bainite) and Equation (18) for martensite transformation. The start and final temperatures and times of phases are defined using an interpolated CCT diagram (Figure 13), whereas the final fractions of phases after cooling to ambient temperature are determined using microscopically analyzed volume fractions of microstructure constituents (Figure 12). The austenite fraction during cooling is defined using Equation (16), along with the rest of the volumetric fractions of phases, resulting in 100% of the entire microstructure (1 in the volume fractions).

The solid lines in the dilatometric curves presented on the left side of Figure 19, Figure 20 and Figure 21 represent the isotropic strains calculated using Equation (20) and the CCT diagram for the cooling process (Figure 13), as well as the volume fractions of phases calculated using Equations (15)–(19). The dashed lines provide experimentally measured isotropic strains in the steel samples. The thermal expansion coefficients and structural strains presented in Table 5 are the result of tuning the model by changing these parameters in order to achieve convergence between the calculation results and the measurement results. 

#### 5.2.2. Prediction of Structural Composition

The kinetics of phase transformations in the solid state, calculated for the entire cross-section of the weld, allow for the determination of the final fraction of phases after cooling to ambient temperature. The field of final fractions of ferrite, bainite, and martensite for thermal cycles obtained from simulations of the temperature field of hybrid welding using an interpolated model of a laser beam and beam focusing at the top surface of the workpiece is presented in Figure 22, whereas the phase fractions of phases determined for beam focusing at z = 5 mm are presented in Figure 23.

## 6. Conclusions

Welding is a sophisticated technological process accompanied by many physical phenomena, especially when using a laser beam as a heat source to penetrate the material. There is a noticeable difference between the distribution of the Gaussian-like TEM_00_ laser beam and the Yb:YAG solid-state laser, which is shown in our experimental research (see Table 1). The aim of developing the interpolated model was to accurately reproduce the power distribution of the Yb:YAG laser beam based on experimental research conducted on the analysis of the energy profile of the solid-state laser beam. Figure 9 shows the differences between the standard Gaussian distribution (Figure 9a), the Super-Gaussian distribution (Figure 9b), the interpolated model (Figure 9c), and the results of the measurement. As it can be observed, the differences between the distribution of the standard Gaussian model and the experimental results are significant, up to 34%. The Super-Gaussian distribution corresponds with the results of the measurement on the *x*-axis or the *y*-axis, but the characteristics of the distribution have changed from a plane to a non-circle distribution. This non-circle distribution does not correspond to the real distribution of the laser beam (see Table 1). Moreover, none of the analytical models allows for the mapping of local extrema existing in the distribution of this laser beam. It is desirable to use the experimental data in numerical modeling to properly map the real energetic conditions of the process. Therefore, the results of the measurements of the Yb:YAG laser beam profile are used in this work in the interpolated heat source model.

The results of real welded tests, as well as computer simulations of thermal phenomena in the hybrid welding process, show that beam focusing has a significant impact on the generated heat input and, in consequence, on the temperature profile of the welded joint (see Figure 14, Figure 15, Figure 16 and Figure 17). The numerically predicted results for the FZ and the HAZ agree well with the results of our experiment (see Figure 18), which proves the correctness of the approach adopted for the numerical analysis of physical phenomena in the hybrid welding process.

The use of interpolation functions and the results of our experimental research (dilatometric analysis) allowed us to describe the CCT diagram and implement the model of phase transformations in the solid state in terms of the kinetics of phase transformations and isotropic strains (see Figure 13 and Table 3).

From the comparison of the calculated final fractions of phases during cooling (see Figure 19, Figure 20 and Figure 21), it can be observed that the model agrees well with the experimentally determined final fractions of phases. The tuning process allowed us to define the proper thermal expansion coefficients and proper structural strains for each phase.

The selection of the thermal load model in the analysis, as well as beam focusing, had an impact on the determined structure composition. There was a visible difference between the structural compositions determined for beam focusing at z = 0 and z = 5 mm (see Figure 22 and Figure 23).

The developed approach can be a useful tool for predicting the basic parameters of the hybrid welding process. The developed computer solvers can hold practical significance for welding technologists.

## Figures and Tables

**Figure 1 materials-17-02364-f001:**
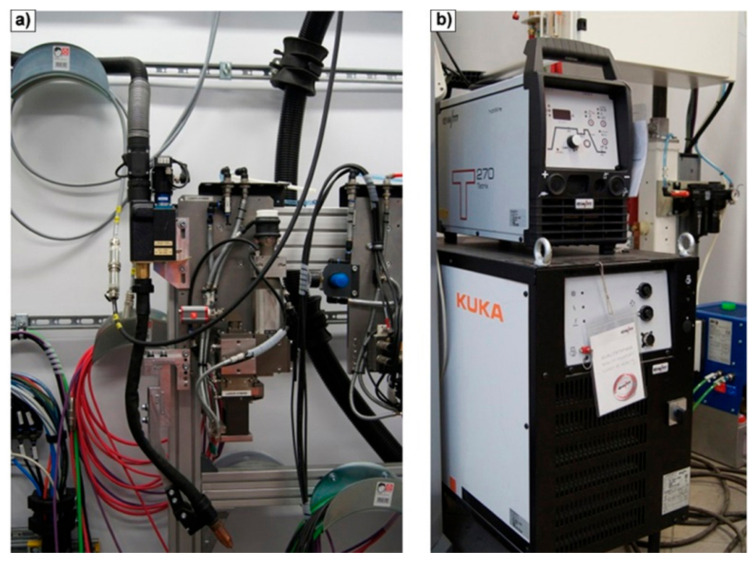
Hybrid welding station: (**a**) laser head with arc torch, (**b**) current source for GMAW.

**Figure 2 materials-17-02364-f002:**
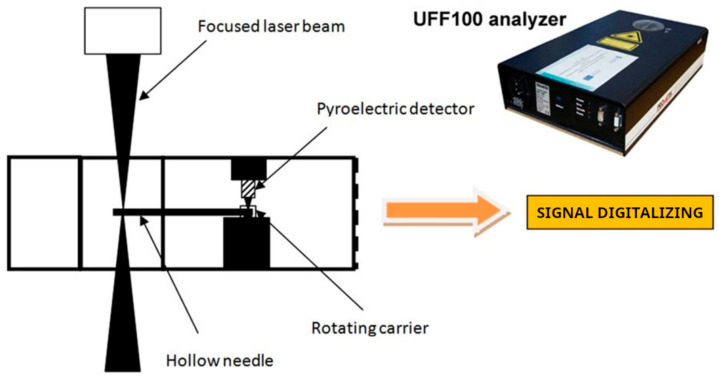
Prometec UFF100 laser beam analyzer—sketch on the measurement of laser beam profile.

**Figure 3 materials-17-02364-f003:**
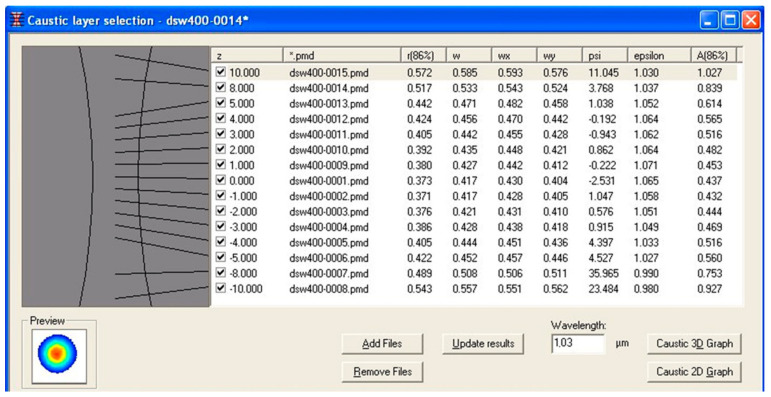
Laser beam power distribution and beam caustic visualizations in analyzer software UFF100 v.1.0.

**Figure 4 materials-17-02364-f004:**
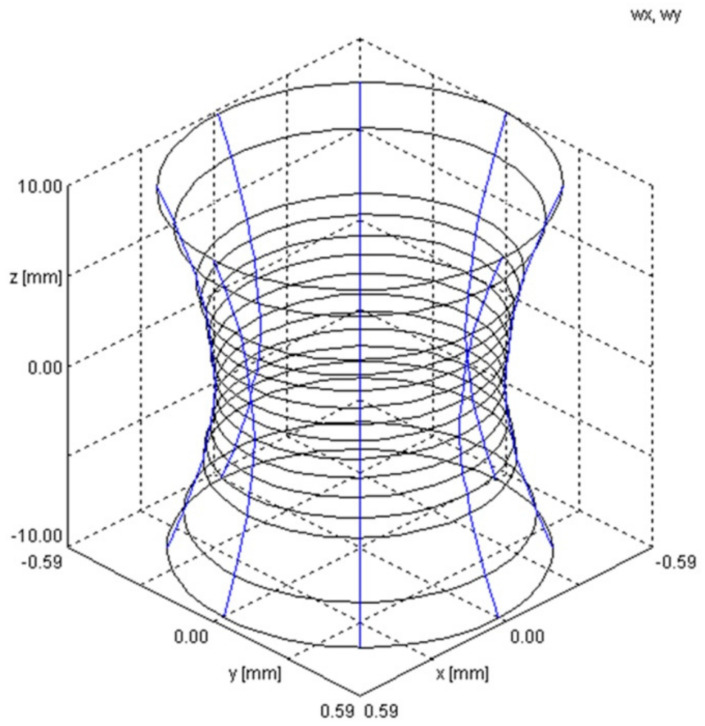
Determined laser beam caustics.

**Figure 5 materials-17-02364-f005:**
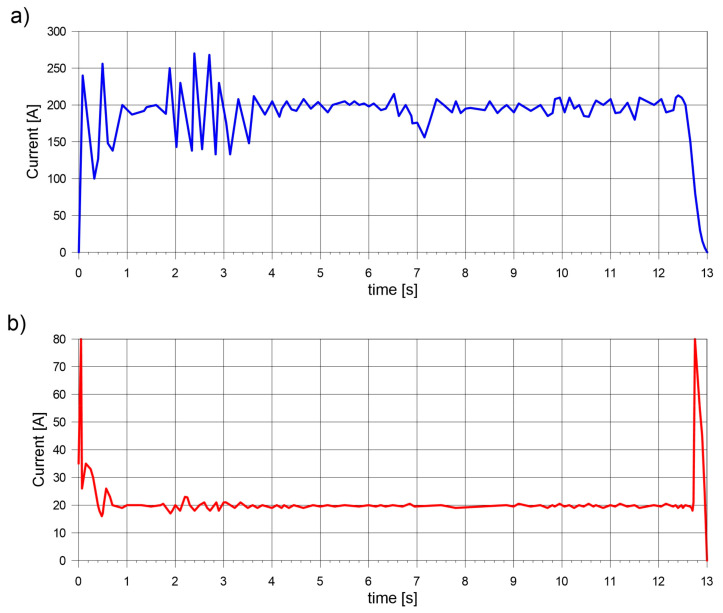
Measured (**a**) current and (**b**) voltage of electric arc.

**Figure 6 materials-17-02364-f006:**
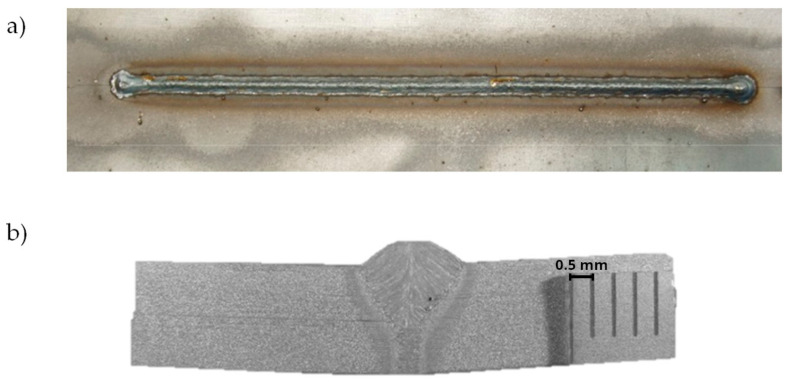
Macroscopic picture of hybrid welded joint made for focusing at z = 0: (**a**) face of the weld and (**b**) cross-section of the weld.

**Figure 7 materials-17-02364-f007:**
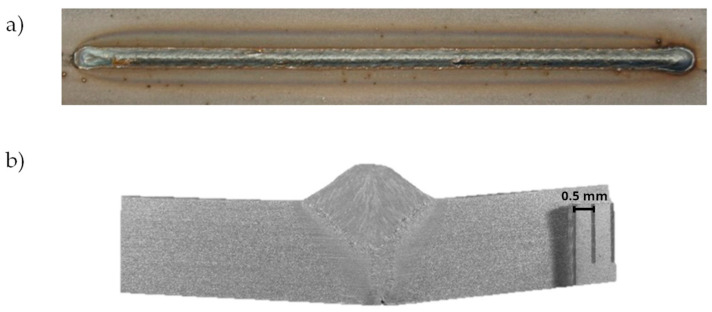
Macroscopic picture of hybrid welded joint made for focusing at z = 5 mm: (**a**) face of the weld and (**b**) cross-section of the weld.

**Figure 8 materials-17-02364-f008:**
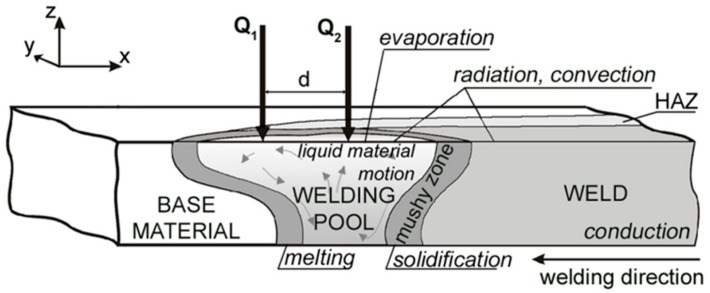
Scheme of phenomena in hybrid welding process.

**Figure 9 materials-17-02364-f009:**
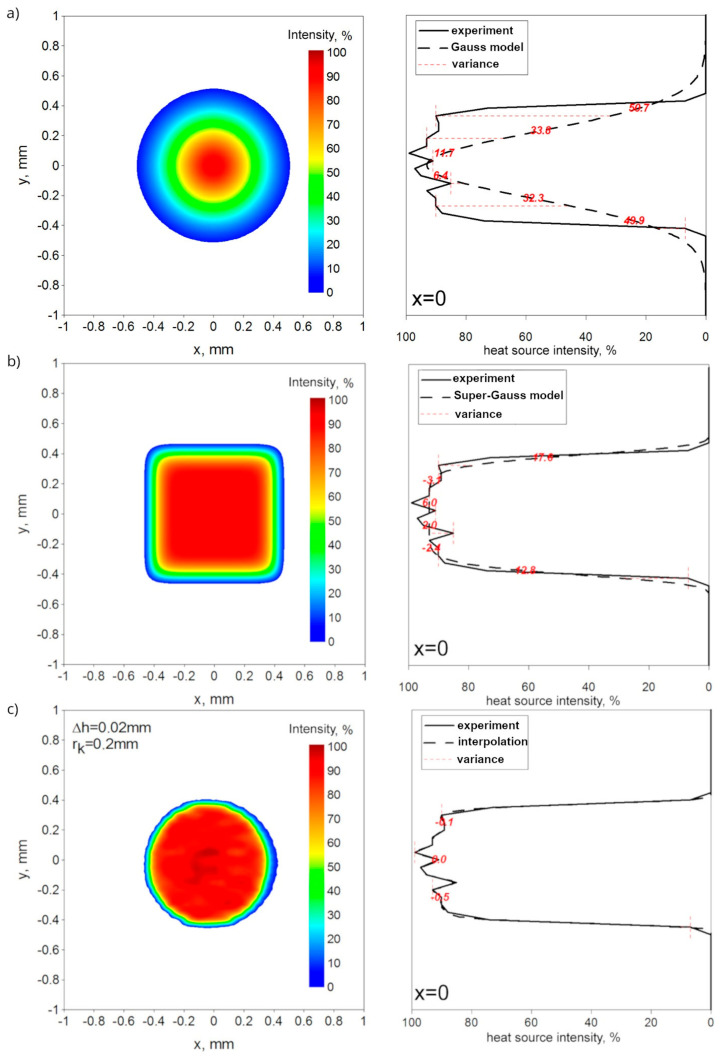
Distribution of laser beam described by (**a**) Gauss model, (**b**) Super-Gauss model, and (**c**) interpolation model for focusing at z = 0 (as percentages).

**Figure 10 materials-17-02364-f010:**
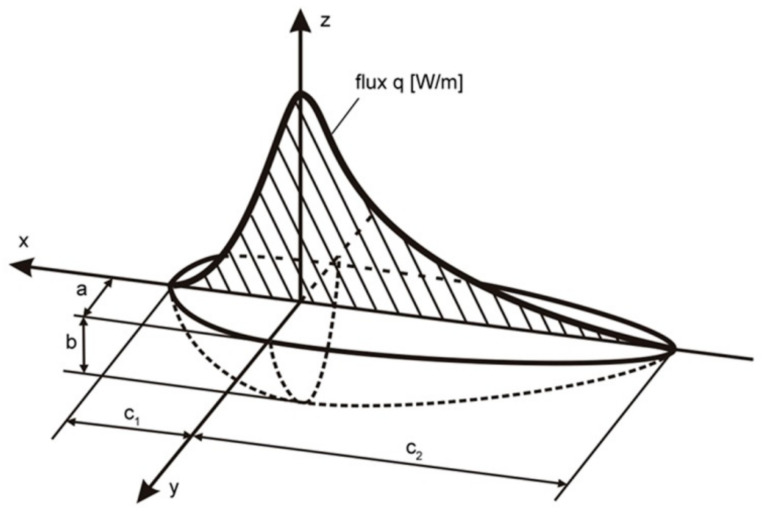
Goldak’s volumetric heat source shape.

**Figure 11 materials-17-02364-f011:**
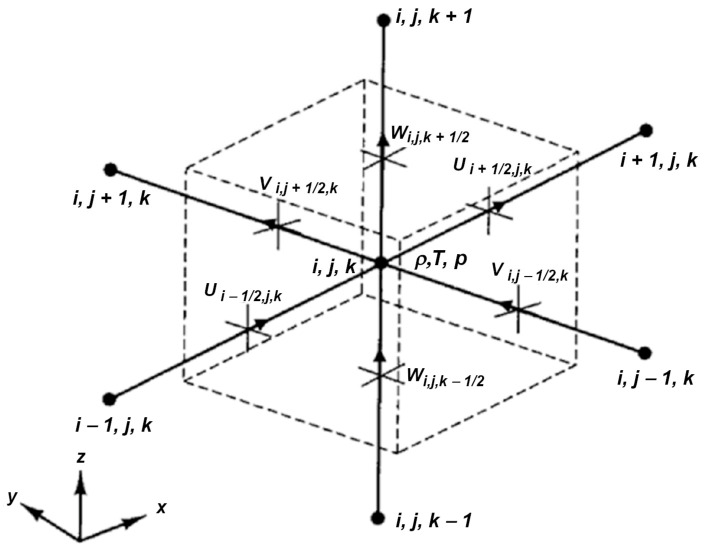
Control volume with marked simulated properties.

**Figure 12 materials-17-02364-f012:**
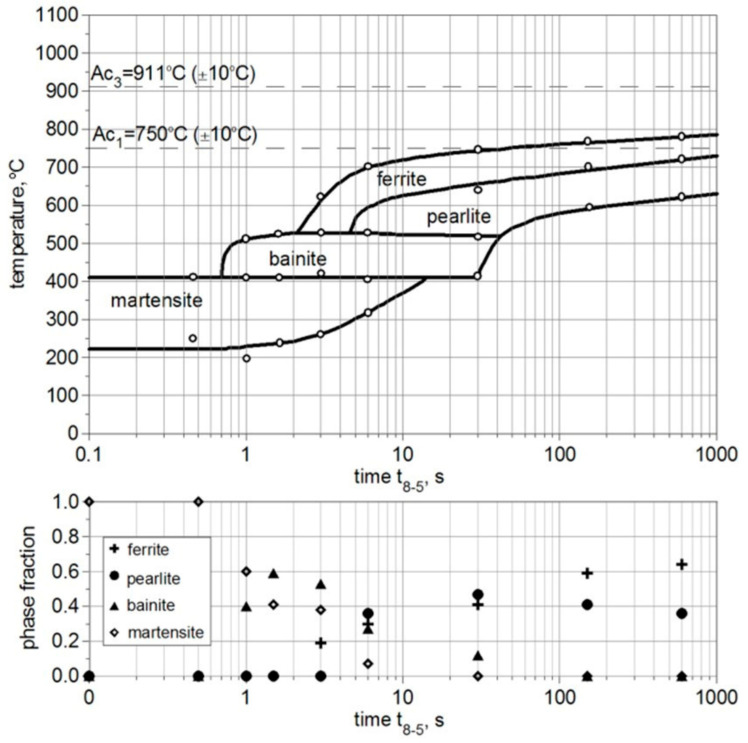
Experimentally obtained CCT diagram with volume fractions of microstructure constituents.

**Figure 13 materials-17-02364-f013:**
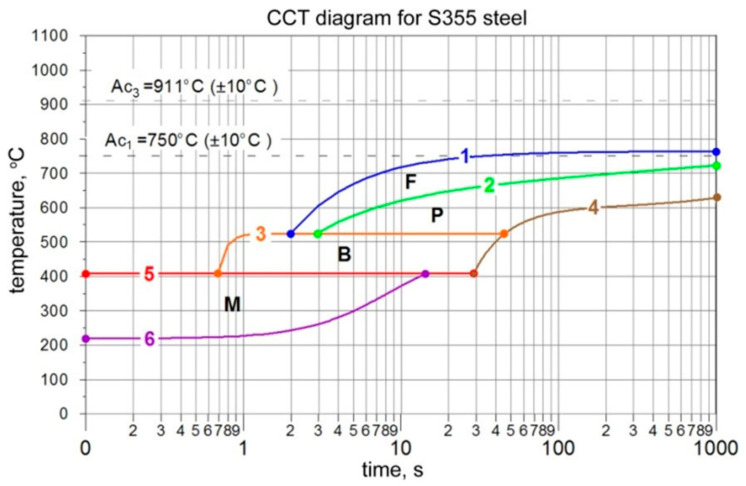
Interpolated CCT diagram for S355 steel.

**Figure 14 materials-17-02364-f014:**
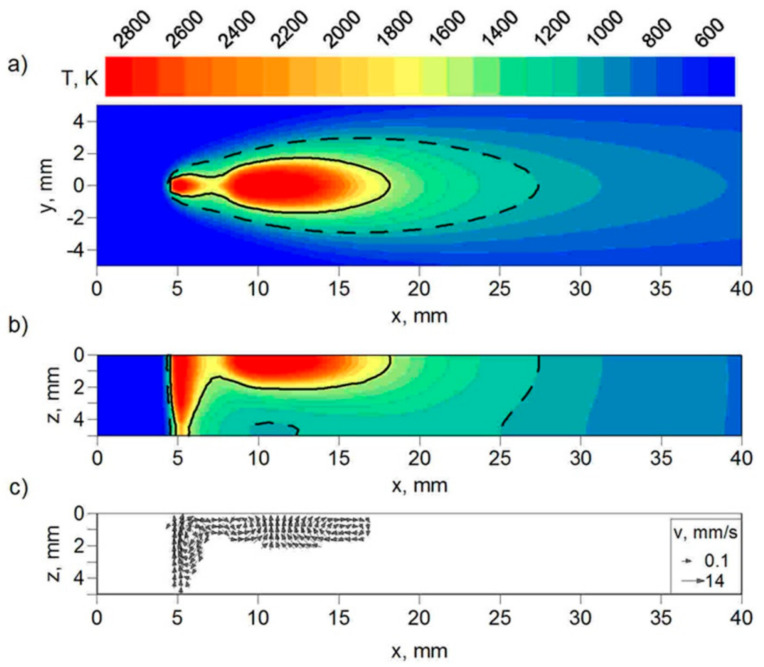
Field of temperature (**a**) at the top surface (z = 0), (**b**) in longitudinal section (y = 0) of the joint, and (**c**) melted material velocity field in longitudinal section (y = 0). Laser beam focusing: z = 0.

**Figure 15 materials-17-02364-f015:**
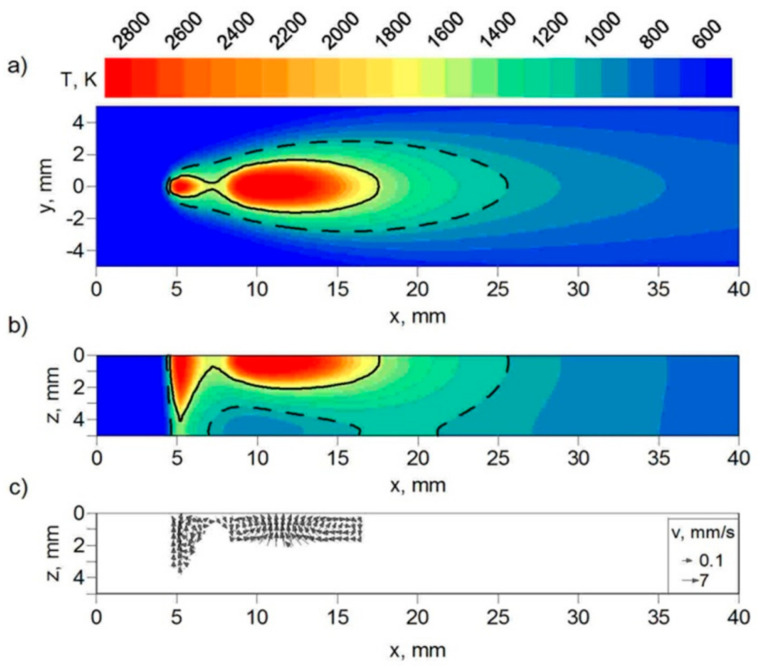
Field of temperature (**a**) at the top surface (z = 0), (**b**) in longitudinal section (y = 0) of the joint, and (**c**) melted material velocity field in longitudinal section (y = 0). Laser beam focusing: z = 5 mm.

**Figure 16 materials-17-02364-f016:**
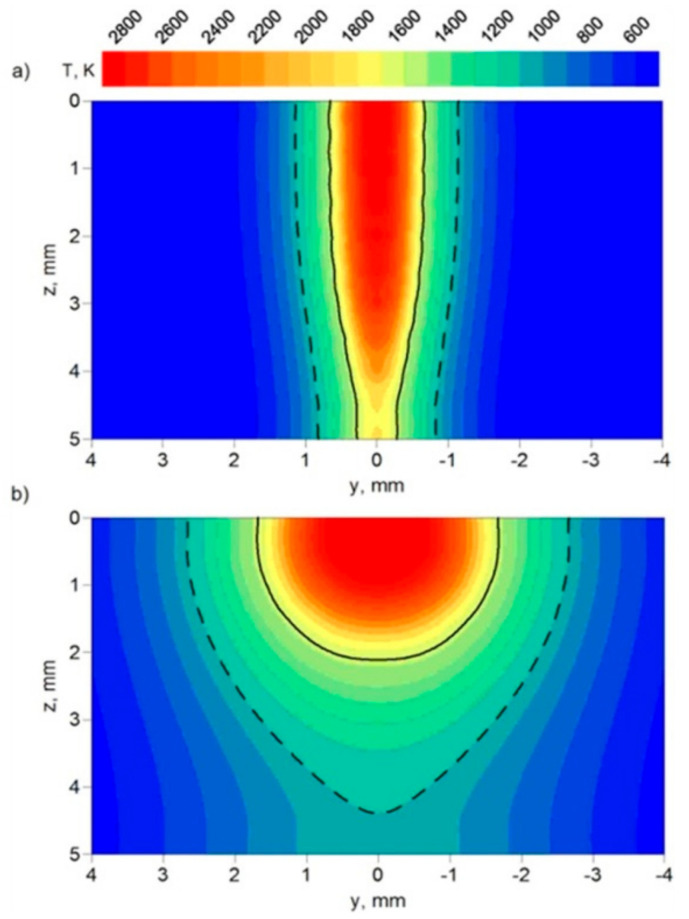
Temperature distribution in the cross-section of the joint for (**a**) x = 5.4 mm and (**b**) x = 12 mm along the welding line. Beam focusing: z = 0.

**Figure 17 materials-17-02364-f017:**
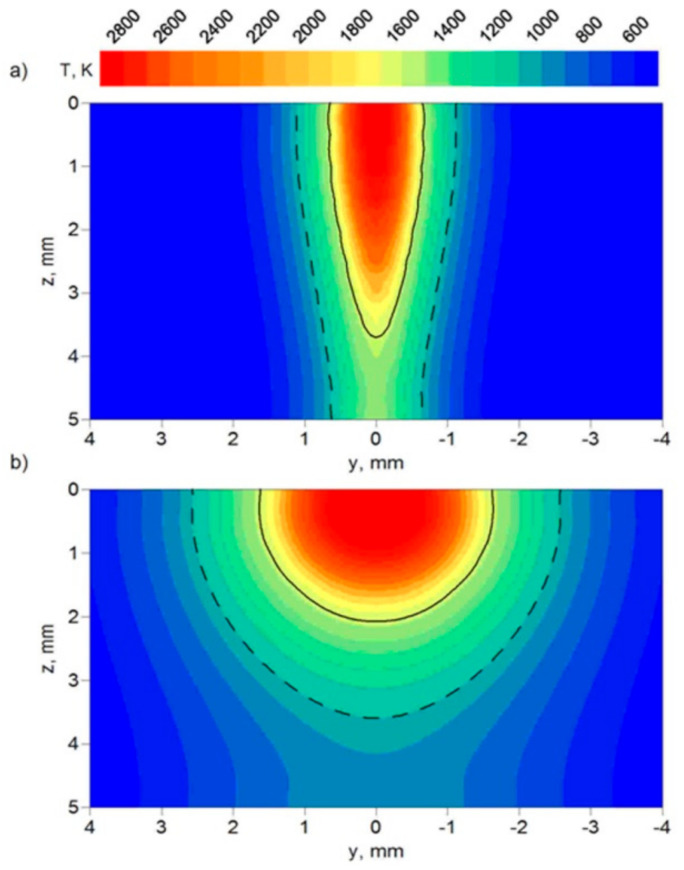
Temperature distribution in the cross-section of the joint for (**a**) x = 5.4 mm and (**b**) x = 12 mm along the welding line. Beam focusing: z = 5 mm.

**Figure 18 materials-17-02364-f018:**
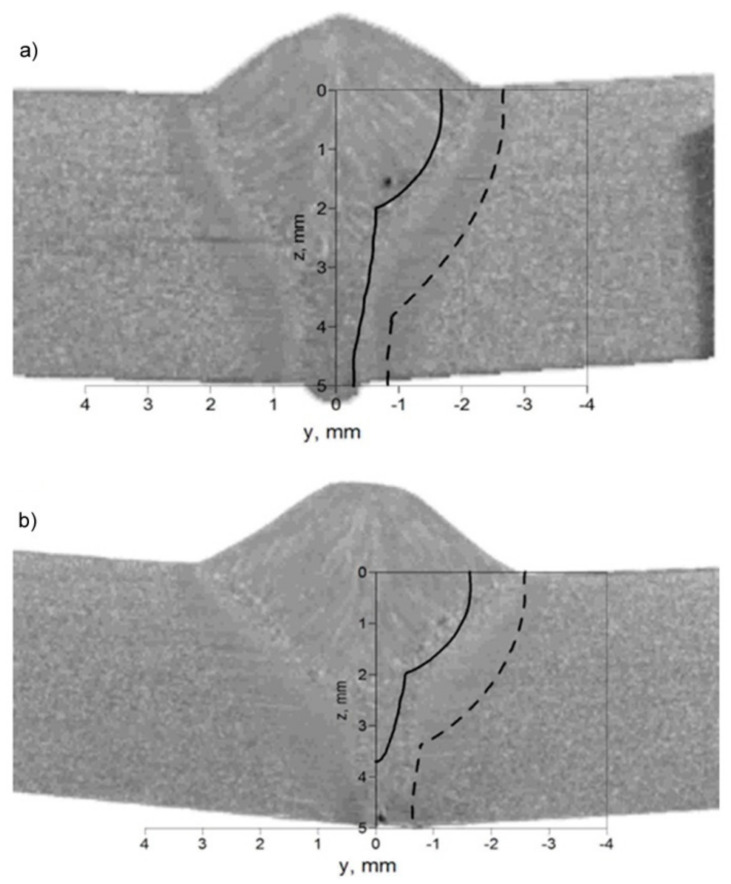
Comparison of predicted FZ (solid line) and HAZ (dashed line) with macroscopic picture of the cross-section of the weld. Beam focusing: (**a**) z = 0 and (**b**) z = 5 mm.

**Figure 19 materials-17-02364-f019:**
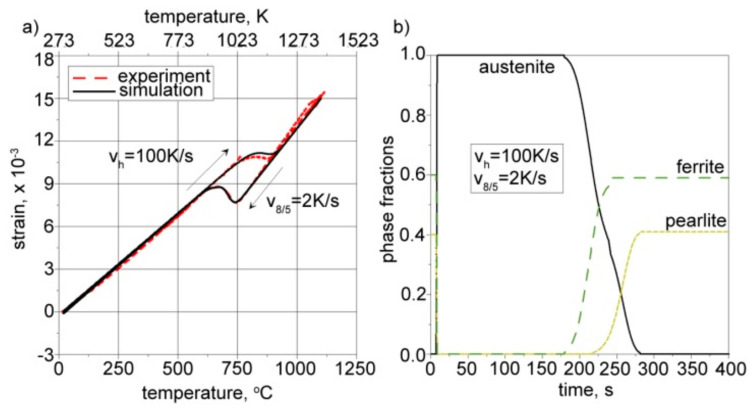
Calculated (**a**) isotropic strain and (**b**) corresponding kinetics of phase transformations. Cooling rate: *v*_8/5_ = 2 K/s.

**Figure 20 materials-17-02364-f020:**
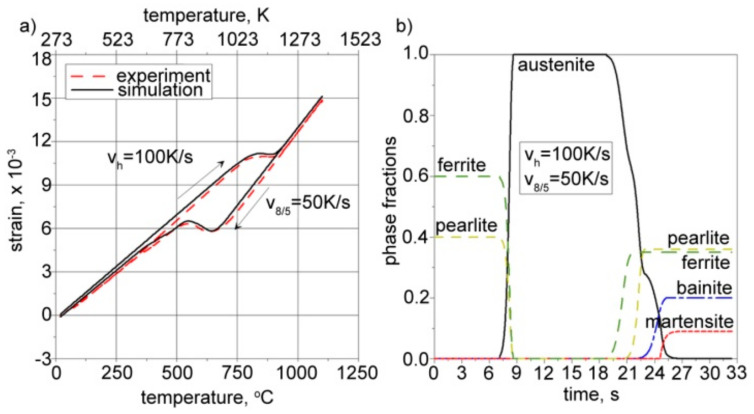
Calculated (**a**) isotropic strain and (**b**) corresponding kinetics of phase transformations. Cooling rate: *v*_8/5_ = 50 K/s.

**Figure 21 materials-17-02364-f021:**
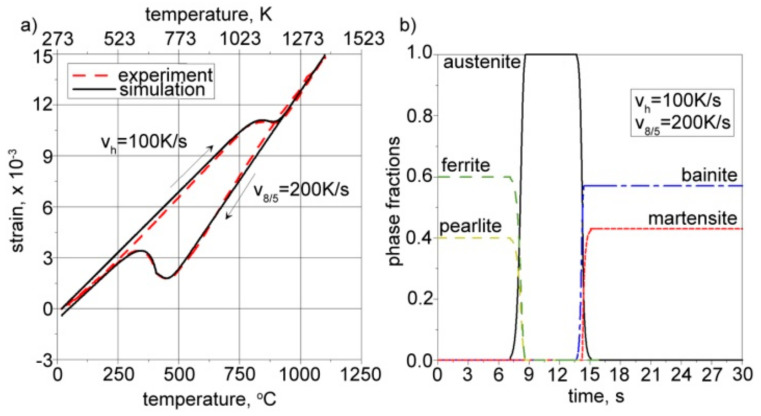
Calculated (**a**) isotropic strain and (**b**) corresponding kinetics of phase transformations. Cooling rate: *v*_8/5_ = 200 K/s.

**Figure 22 materials-17-02364-f022:**
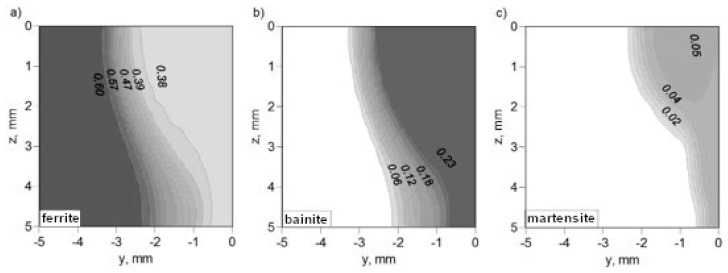
Prediction of structural composition of welded joint for beam focusing at z = 0: (**a**) volume fractions of ferrite, (**b**) volume fractions of bainite and (**c**) volume fractions of martensite.

**Figure 23 materials-17-02364-f023:**
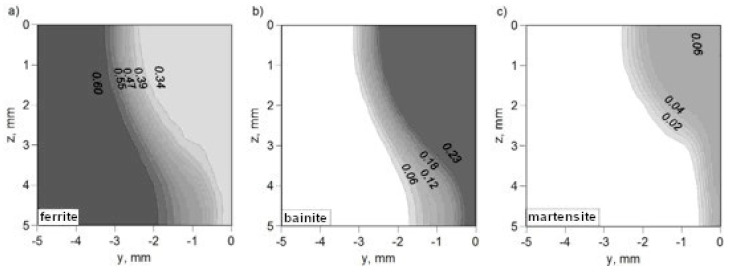
Prediction of structural composition of welded joint for beam focusing at z = 5 mm: (**a**) volume fractions of ferrite, (**b**) volume fractions of bainite and (**c**) volume fractions of martensite.

**Table 1 materials-17-02364-t001:** Measured distribution of the Yb:YAG laser power.

Beam Focusing	Diagram of Power Density Distribution
z = 5 mm	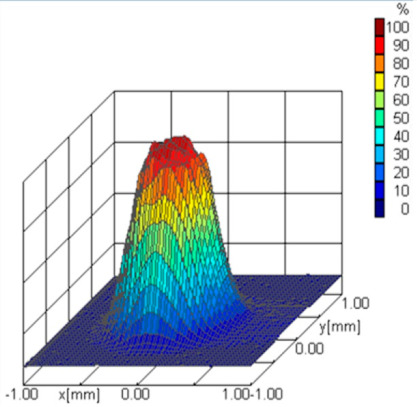
z = 0	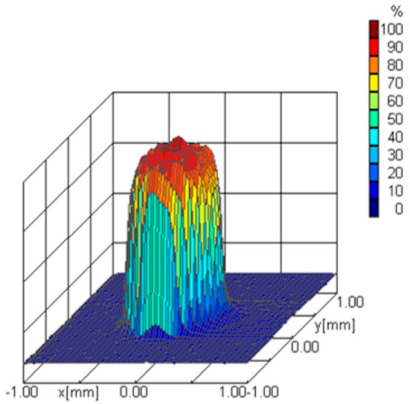

**Table 2 materials-17-02364-t002:** Hybrid laser–arc welding process parameters.

Welding Speed	Laser-to-Arc Distance	MIG/MAG	Laser (Leading Source)
v = 1 m/min	*d* ≈ 3 mm	Energetic param.: *I* = 190 A, *U* = 19 V Shielding gas: 82%Ar + 18%CO_2_, 18 L/min Feed rate: 6 m/min Torch angle: 20° CTWD: ~20 mm	Q_L_ = 3000 W

**Table 3 materials-17-02364-t003:** Interpolated functions determining the CCT diagram.

Curve (Figure 13)	Time	Equation	Coefficients
1 *F_s_*	1.98–1000	$y=A_{Fs}+\frac{B_{Fs}}{t}$	*A_Fs_* = 764.38 *B_Fs_* = −473.38
2 *P_s_*	2.90–1000	$y=A_{Ps}\cdot B_{Ps}\frac{1}{t}\cdot t^{C_{Ps}}$	*A_Ps_* = 625.49 *B_Ps_* = 0.56 *C_Ps_* = 0.02
3 *B_s_*	0.69–46	$y=\frac{A_{Bs}\cdot B_{Bs}+C_{Bs}\cdot t^{D_{Bs}}}{B_{Bs}+t_{D_{Bs}}}$	*A_Bs_* = −745.97 *B_Bs_* = 0.0034 *C_Bs_* = 523.30 *D_Bs_* = 8.92
4 *P_f_ + B_f_*	29.10–1000	$\begin{array}{l} y=A_{Pf+Bf}+\\ +B_{Pf+Bf}\cdot t+\frac{C_{Pf+Bf}}{2}t \end{array}$	*A_Pf+Bf_* = 600.22 *B_Pf+Bf_* = 0.03 *C_Pf+Bf_* = −161,246.52
5 *M_s_*	0.10–29.10	$y=A_{Ms}+B_{Ms}\cdot t$	*A_Ms_* = 409 *B_Ms_* = 0
6 *M_f_*	0.10–14.50	$y=\frac{A_{Ms}\cdot B_{Ms}+C_{Ms}\cdot t^{D_{Ms}}}{B_{Ms}+t^{D_{Ms}}}$	*A_Mf_* = 219.43 *B_Mf_* = 32.36 *C_Mf_* = 484.25 *D_Mf_* 1.64

**Table 4 materials-17-02364-t004:** Thermo-physical parameters used in computer simulations.

Nomenclature	Symbol	Value
Solidus temperature	*T_S_*	1750 K
Liquidus temperature	*T_L_*	1800 K
Boiling point	*T_b_*	3010 K
Ambient temperature	*T* _0_	293 K
Specific heat of solid phase	*c_S_*	650 J kg^−1^K^−1^
Specific heat of liquid phase	*c_all_*	840 J kg^−1^K^−1^
Density of solid phase	*ρ_S_*	7800 kg m^−3^
Density of liquid phase	*ρ_L_*	6800 kg m^−3^
Latent heat of fusion	*H_L_*	270 × 10^3^ J kg^−1^
Latent heat of evaporation	*H_b_*	76 × 10^5^ J kg^−1^
Thermal conductivity of solid phase	*λ_S_*	45 W m^−1^K^−1^
Thermal conductivity of liquid phase	*λ_L_*	35 W m^−1^K^−1^
Convective heat transfer coefficient	*α*	50 W m^−2^K^−1^
Boltzmann’s constant	*σ*	5.67 × 10^−8^ W m^−2^K^−4^
Thermal expansion coefficient	*β_T_*	4.95 × 10^−5^ K^−1^
Surface radiation emissivity	*ε*	0.5
Dynamic viscosity	*μ*	0.006 kg m^−1^s^−1^
Solid-particle average diameter	*d* _0_	0.0001 m

**Table 5 materials-17-02364-t005:** Thermal expansion coefficients and structural strains of micro-constituents.

Structural Constituent	Thermal Expansion Coeff. *α_(i)_*, ×10^−6^, 1/°C	Structural Strains *ε_(i)_*, ×10^−3^
austenite	*α_A_ =* 21.0	*ε_A_ =* 3.5
ferrite	*α_F_ =* 14.7	*ε_F_ =* 3.0
pearlite	*α_P_ =* 13.7	*ε_P_ =* 4.0
bainite	*α_B_ =* 12.5	*ε_B_ =* 3.5
martensite	*α_M_ =* 12.0	*ε_M_ =* 5.7

## Data Availability

The original contributions presented in the study are included in the article, further inquiries can be directed to the corresponding authors.

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
