# Peer review of "Modeling of Yb:YAG Laser Beam Caustics and Thermal Phenomena in Laser–Arc Hybrid Welding Process with Phase Transformations in the Solid State"

_materials, 2024, doi:10.3390/ma17102364_

Round 1

Reviewer 1 Report

Comments and Suggestions for Authors

+ The abstract require to solve the following questions: What problem did you study and why is it important? What methods did you use? What were your main results? And what did you conclude from your results? Please make your abstract with specific and quantitative results to further enrich the content of the article. Please indicate each of these questions in the abstract separately when you are replying to this comment. 

+ Please in line 30, is CO2

+ At the end of the introduction (final paragraph). The authors need to clarify the novelties of this work. The authors should revise the Introduction substantially.

+ The methodology, Methods, and Materials section must give a clear overview of what was done and give enough information to replicate the study (like a recipe!); be complete, but make life easy for your reader! break into smaller sections with subheadings, cite references for commonly used methods, and display a flow diagram or data table where possible.

+ From materials and methods: Is any standard used for the welding laser process?

+ Fig. 5 requires improvements, its hard to see. Check author guidelines.

+ How table 1 was determined? Any previous work?

+ From figs 6 and 7 its required an scale bar.

+ From equations used, please provide a reference.

+ The authors mention that: visible difference in power distribution mapping. Then why this is possible with the geometry change? I note that heat source curves look similar.

+ Results: The explanation of the figs needs to appear before the figure.

From FEM results, could be discussed in a depth way and compare with other works. Then which software was used?

+ A symbology and acronyms list its required.

+ The Conclusions could be rewritten according to the activities developed in a depth way. You can use percentages, main findings, values, etc.

Comments on the Quality of English Language

minor grammar issues detected

Author Response

Dear Reviewer,

Thank you very much for the review and your valuable comments. Please find enclosed in the attachment responses to your comments with an indication of the corrected text fragments in the paper.
Once again thank you very much for the review. We hope that revised version of the paper will be suitable for printing in the journal.

Best regards,
Authors

Reviewer 2 Report

Comments and Suggestions for Authors

The presented research is an interesting topic and authors did their best to present results of voluminous experimental investigation and numerical simulations. Results and discussion seem impressive and do not require extensive corrections or further improvements.

- The only point that appears puzzling is discussion on comparison between the analytical Gaussian-like laser power distribution and the real distribution obtained experimentally (page 10 – the third paragraph). First of all, the remark “and proved in prior work [18] (by almost all the same authors!!) is not plausible in this context. Authors should have repeated those proofs, even in an abbreviated form, and not direct the reader to their previous publication. Furthermore, if “there is a visible difference in power distribution mapping between analytical models and interpolated model”, why insist on using the interpolated model?

- The paper is well written, though there are some problems with English language grammar - please consult the enclosed scanned pages of the manuscript with proposed corrections.

- The scanned pages of the manuscript with marked errors and proposed corrections are enclosed.

Comments on the Quality of English Language

There should be a space between some variables value and its units.

Some ususal phrases are written in a wrong way (e.g. "on the basis: instead of "based on", etc.), since the authors obviously did not perform the check spelling.

There are a few misprints,as well.

Author Response

(The authors gave the same response as above.)

Round 2

Reviewer 1 Report

Comments and Suggestions for Authors

The authors have made an effort to improve the quality of the manuscript, but some minor issues need to be solved before publication:

+ Please provide a plot, not a photo of Figure 5.

+ The scale bar from Figures 6 and 7 is not clear. Please use lines or some similar ones to identify properly.

+ Please from Figs 19 to 20. Provide an explanation of the phase transformation of the material (austenite, ferrite, pearlite, and bainite) and how this was identified in time and fractions.

Author Response

Dear Reviewer,

once again thank you very much for the review and remarks. In the attachment please find respond to your comments with an indication of lines with added new explanations and improved figures in the paper.

We hope that revised version of the paper will be suitable for printing in the journal.

Best regards,
Authors
